# Research Progress on the Construction and Application of a Diabetic Zebrafish Model

**DOI:** 10.3390/ijms24065195

**Published:** 2023-03-08

**Authors:** Yu Cao, Qianqian Chen, Yinai Liu, Libo Jin, Renyi Peng

**Affiliations:** Institute of Life Sciences & Biomedicine, Collaborative Innovation Center of Zhejiang Province, College of Life and Environmental Science, Wenzhou University, Wenzhou 325035, China; 22461338001@stu.wzu.edu.cn (Y.C.); 21451335004@stu.wzu.edu.cn (Q.C.); 21461338012@stu.wzu.edu.cn (Y.L.)

**Keywords:** diabetes, zebrafish, inflammation, animal models, complications

## Abstract

Diabetes is a metabolic disease characterized by high blood glucose levels. With economic development and lifestyle changes, the prevalence of diabetes is increasing yearly. Thus, it has become an increasingly serious public health problem in countries around the world. The etiology of diabetes is complex, and its pathogenic mechanisms are not completely clear. The use of diabetic animal models is helpful in the study of the pathogenesis of diabetes and the development of drugs. The emerging vertebrate model of zebrafish has many advantages, such as its small size, large number of eggs, short growth cycle, simple cultivation of adult fish, and effective improvement of experimental efficiency. Thus, this model is highly suitable for research as an animal model of diabetes. This review not only summarizes the advantages of zebrafish as a diabetes model, but also summarizes the construction methods and challenges of zebrafish models of type 1 diabetes, type 2 diabetes, and diabetes complications. This study provides valuable reference information for further study of the pathological mechanisms of diabetes and the research and development of new related therapeutic drugs.

## 1. Introduction

Diabetes mellitus, mainly type 1 diabetes mellitus (T1DM) and type 2 diabetes mellitus (T2DM), is a group of metabolic diseases characterized by chronic hyperglycemia with multiple causes [1]. A long-term hyperglycemic state in patients with diabetes can cause chronic damage and the dysfunction of various tissues of the body, especially the eyes, kidneys, heart, blood vessels, and nerves, which seriously endangers human health. Diabetes has become a worldwide public health problem that seriously threatens human health. Diabetes is a global epidemic, and its prevalence rate increases each year [2,3,4,5]. Thus, it has become an important cause of death and disability. According to the latest statistics from the International Diabetes Federation, the total number of people with diabetes in the world has reached 537 million, of which 6.7 million people have died from diabetes. By 2045, the number of people with diabetes will increase by 73.6 million, and the number of people who have died from diabetes will increase by 2.5 million. At present, the study of diabetes is an important topic worldwide, but its etiology has not yet been fully clarified [6,7,8,9].

Animal models can reflect pathophysiological changes in human diseases, and can be used to study disease mechanisms and screen active drugs. At present, a large number of studies using animal models of diabetes to research diabetes and its complications have been reported (Figure 1). Among these studies, the most commonly used animal models were rats, mice, pigs, and monkeys. Each animal model has its own advantages and disadvantages, but no animal model can fully represent the clinical symptoms of human diabetes [10,11,12,13,14]. In the field of scientific research, the zebrafish model has gradually attracted the attention of many researchers due to its many advantages that other model animals do not have, and it has become a very important vertebrate model organism (Figure 1). For example, zebrafish are easy to breed because of their small size, large number of eggs, short growth cycle, developmental transparency, and simple adult fish breeding, which can effectively improve experimental efficiency and is suitable for large-scale experimental studies [15,16,17]. Additionally, the genetic information of zebrafish is well documented, and the gene manipulation technology for zebrafish is very advanced. By the targeted induction or knockout of certain genes in the body and changing the expression level of these genes, some human disease states can be easily replicated in zebrafish (Table 1) [18,19]. Thus, zebrafish are highly suitable for human disease modeling and mechanistic research. The zebrafish model has its own characteristics and advantages in terms of endocrine gland function and glucose metabolism, and this model has been applied to the study of human diabetes and other metabolic diseases [20].

Various developmental stages of zebrafish can be found that contrast with the development process of mammals. Furthermore, zebrafish only need 72 h to develop from fertilized eggs to hatching, which is much shorter than the development time of mammals such as mice and humans (Figure 2) [30,31,32,33]. Zebrafish embryo development is divided into seven stages—zygote, cleavage, blastula, gastrum, segmental, pharyngeal sac, and incubation. Among them, the incubation stage is very important. As the last period of embryonic development of zebrafish, embryos in the incubation stage still continue to grow at the previous speed, and the morphogenesis of many organ primordia comes to an end and slows down somewhat. Zebrafish embryos develop outside of the mother, and the entire development process is transparent, which is conducive to real-time observation and detection. Therefore, the early developmental morphology of zebrafish can be used to accurately and efficiently judge the toxicity of embryos exposed to drugs. This process is widely used in the study of drugs’ developmental toxicity and teratogenicity. In zebrafish, the major organs related to metabolic regulation (e.g., pancreas) and some insulin-sensitive peripheral tissue organs (e.g., liver, muscle, etc.) are evolutionarily conserved. The cellular structure of the zebrafish pancreas is similar to that in humans. The zebrafish pancreas is composed of an exocrine gland (glandular duct structure) and an endocrine gland; the endocrine gland is composed of α, β, δ, and PP cells. These cells are able to express and produce glucagon, insulin, somatostatin, and pancreatic polypeptides, respectively [34,35,36]. In the first week of embryonic development, the zebrafish pancreas is simple, with only one islet. In adult zebrafish, some pancreatic beta cells gather at the edge of the head of the pancreas, forming a single large islet. However, others form small islets which are embedded in the exocrine gland tissue and dispersed along the pancreatic duct. The zebrafish pancreas originates from two bud bases, the dorsal bud and the abdominal bud. The dorsal bud forms 24 h after fertilization and mainly develops into the pancreatic endocrine gland, namely, the primary islet. Thirty-four hours after fertilization, the abdominal buds form and gradually fuse with the dorsal buds to form the pancreas. The abdominal buds differentiate into pancreatic endocrine glands and pancreatic exocrine glands. Important genes and signaling pathways regulating endocrine pancreas development in zebrafish are consistent with those in other vertebrates. Five days after fertilization, the basic structure and function of the zebrafish pancreas are highly similar to those of the adult mammalian pancreas. The zebrafish pancreas is sensitive to insulin, and other tissues and organs involved in blood glucose regulation (liver, muscle, adipose tissue, etc.), as well as some of the molecular mechanisms related to the regulation of glucose metabolism, are evolutionally consistent with those of mammals [37].

In addition, glucose metabolism is an important part of whole-body metabolism in zebrafish. Studies investigating zebrafish carbohydrate utilization have supported this notion. First, zebrafish are omnivorous teleosts that live in areas with warm water. The carbohydrate utilization rate and glucose metabolism of carnivorous teleosts were lower than those of zebrafish [38]. Second, carbohydrates are an important component of the zebrafish diet, and the growth rate of zebrafish is proportional to the carbohydrate content. Third, some important genes related to glucose metabolism, such as hexokinase genes and glucose transporter genes, are adequately expressed in zebrafish. In contrast, the deletion of these genes causes a series of severe neurological defects in zebrafish embryos. Fourth, due to partial genome duplications, many genes in zebrafish contain two corresponding orthologous genes. These genes contain different enhancers and are functionally expressed in different tissues, such as the *nkx2.2a*, *pax6b*, and *jagged1b* genes in the pancreas.

Taking advantage of the characteristics of zebrafish gene expression, a large number of zebrafish mutants and transgenic zebrafish lines have been constructed. For example, green fluorescent transgenic zebrafish can specifically express green fluorescent proteins in the organs or tissue cells of the zebrafish body. With the transparency that is characteristic of zebrafish larvae, changes in organ morphology and tissue composition can be directly observed under a microscope. This can effectively simplify the experimental process and is highly suitable for high-throughput screening of drugs. These characteristics make the zebrafish a good model for studying the regulation of glucose metabolism [39,40,41,42,43,44]. In addition, a zebrafish angiogenesis model carrying the loss of function mutation of the threonyl-trna synthetase (tars) gene was constructed. This transgenic fish could easily demonstrate the branching and formation of blood vessels in the body. It reflects the specificity and efficiency of multiple stress response pathways that have evolved collectively by ensuring accurate sensing and response to different types of stress for the benefit of the organism, and it can also help us to better understand how regulatory functions are aberrant in diabetic organisms. These properties make zebrafish a good model for studying the regulation of glucose metabolism [45,46,47].

At present, researchers have successfully established different methods for different types of diabetic zebrafish models. T1DM is known as insulin-dependent diabetes mellitus. T1DM is mainly caused by the autoimmune inflammatory response of islet β cells. This response leads to the destruction and dysfunction of β cells, resulting in a lack of insulin secretion. Although an autoimmune-induced model of type 1 diabetes in zebrafish is lacking, a model of type 1 diabetes has been developed that targets damaged β cells [48]. At present, there are four methods to establish a type 1 diabetes model by destroying beta cells: surgical resection, drug induction, genetic ablation, and gene modification [49,50,51,52]. T2DM is induced by a variety of environmental factors, and is characterized by insulin resistance and relative deficiency in insulin secretion. Insulin resistance is considered to be the main driver of type 2 diabetes. T2DM is mainly caused by decreased sensitivity to insulin of its target organs. This condition manifests as impaired glucose utilization by insulin target tissues, such as muscle, liver, and fat tissues, leading to glucose retention in the blood. In the early stages of the disease, the affinity between insulin and its target cells decreases. Thus, β cells are required to secrete and release more insulin, leading to glucagonemia. Thus, a vicious cycle of glucose metabolism disorders and β cell insufficiency is formed. This cycle eventually leads to a severe deficiency in β cell function, and then to T2DM [53,54,55,56]. Animal models of type 2 diabetes include spontaneous animal models (such as *ob/ob* mice), induced animal models (such as high-fat diet and drug intervention experiments), and transgenic models. There are three main methods for the construction of a zebrafish type 2 diabetes model: the glucose immersion method, diet induction method, and gene knockout method [57,58,59,60,61].

Due to the above advantages, zebrafish are an ideal model for the study of diabetes. This review summarizes the methods of the establishment of different types of diabetic zebrafish and the research progress in understanding the pathological mechanisms of diabetes. This study provides valuable reference information for the establishment of a zebrafish diabetes model, the study of the pathogenesis of diabetes, and the high-throughput screening of diabetes treatment drugs.

## 2. Methods of Diabetic Zebrafish Model Construction

### 2.1. Type I Diabetes Model Construction Methods

#### 2.1.1. Method of Surgical Resection

Surgical removal of the pancreas was the earliest method of diabetes replication. As early as 1689, Johann Conrad Brunner recorded polydipsia, polyphagia, and polyuria in dogs after pancreatectomy [62]. More than 200 years later, some scholars have linked the pancreas to diabetes and proved that surgical resection of the pancreas could induce diabetes [63,64,65,66]. Some researchers have tried to remove the pancreas of transgenic zebrafish with green fluorescent protein (GFP)-specifically labeled islets under a light microscope. Because the pancreas of zebrafish is small and diffusely distributed around the liver, gallbladder, and intestine, it is difficult to completely remove the pancreas [45,67,68]. Therefore, the establishment of a zebrafish diabetes model by surgical resection requires highly skilled operating techniques and elaborate equipment for researchers. Furthermore, it is traumatic to zebrafish, resulting in a low survival rate after surgery. Therefore, this model is not often used in zebrafish.

#### 2.1.2. Drug Induction Method

Streptozotocin (STZ) and alloxan are the most commonly used drug-induced diabetes models in rodents. Both induce apoptosis of islet β cells by alkylation or DNA damage, resulting in insufficient insulin secretion and elevated blood glucose levels. However, rodent models generally have the shortcomings of high costs and long cycle times [69,70,71,72], which can be remedied by zebrafish models. STZ is soluble in water, but its aqueous solution is extremely unstable at room temperature. When STZ was injected into adult zebrafish in the study, the zebrafish showed elevated blood glucose levels and a significant reduction in insulin levels in the blood and pancreas [73]. In the streptozotocin-induced diabetic zebrafish model, fasting blood glucose levels were increased, and insulin expression levels were decreased. However, these zebrafish also developed diabetic complications, such as diabetic nephropathy and caudal fin aplasia. Although the construction of the model was successful, these two methods also have shortcomings. For example, after alloxan treatment of fertilized zebrafish embryos, the size of the islets in the embryos decreased, and nearly half of the zebrafish embryos died. Alloxan can also cause liver and kidney damage, and the mortality rate of this model is high, indicating that alloxan is highly toxic to zebrafish embryos. Compared with alloxan, streptozotocin has relatively less tissue toxicity, and researchers established an adult zebrafish model of type 1 diabetes by intraperitoneally injecting streptozotocin at a dose of 350 mg/kg. To maintain the zebrafish in a hyperglycemic state, multiple intraperitoneal injections of streptozotocin are needed, which is cumbersome [74,75].

#### 2.1.3. Induction by Genetic Modification

With the development of transgenic technology, zebrafish can achieve hyperglycemia by targeted gene editing under certain conditions. A study constructed *Tg* (*1.2ins: htBid TE-ON; LR*) using the dual action of tetracycline and ecdysone on the TetOn system and specifically deleted beta cells [76]. This resulted in an absolute insulin deficiency. *Tg* (*1.2ins: Kir6.2CA-GFP TE-ON; Kir6.2*), which encodes the main subunit of the ATP-dependent K^+^ channel on the β cell membrane, induces potassium efflux, hyperpolarizes the cell membrane, and decreases insulin secretion and the compensatory differentiation of β cells through tetracycline [77,78,79]. *Tg* (*ins:loxp:BFPloxp:DTA*) β-cells could specifically express lethal diphtheria toxin α, which is mediated by the transgenic fish *Tg* (*ins:Cre*). Zebrafish larvae that were deficient in β-cells showed higher free glucose levels and significant developmental delays [80,81] in addition to the transgenic zebrafish *Tg* (*HS4-sst2:CFP; ins:PhiYFP-m-dest1-2TA-nfsB*) (*lm lmcoo9*), abbreviated as *Tg* (*ins:NTR*). Green fluorescent proteins and nitroreductase were expressed in the islet β cells of *Tg* (*ins:NTR*) zebrafish. Nitroreductase can convert metronidazole into a cytotoxin and destroy islet β cells. After 3 days of intraperitoneal injection of metronidazole, 99.7% of the β cells were damaged, but the α cells were not damaged. Furthermore, the zebrafish showed hyperglycemia. Five to seven days after metronidazole injection, the number of β-cells began to increase, and the blood glucose levels returned to normal after two weeks, presumably due to the strong regeneration ability of zebrafish. Additionally, the proliferation and differentiation of single cells in the islets led to the regeneration of β-cells. Transgenic technology can be used to generate zebrafish with specific functions, such as green or red fluorescent proteins. However, this could also damage other genes in the genome, leading to the loss of other functional genes, and it takes a long time to obtain homozygotes [82,83,84].

### 2.2. Construction Methods for Type II Diabetes

#### 2.2.1. Glucose Solution Immersion Method

The uptake of glucose by zebrafish is related to osmotic pressure regulation. The hypertonic state in the body causes water from the environment to enter the body, and glucose is then taken into the body. A diabetic zebrafish model can easily be constructed by soaking zebrafish in a solution with an appropriate concentration of glucose. The average fasting blood glucose level of adult zebrafish is 74 ± 8.5 mg/dL, and a level more than three times the normal value is considered hyperglycemic (>200 mg/dL). One study used 2% and 0% glucose solutions alternately for 28 days to establish a zebrafish diabetes model by means of a high glucose exposure method. In addition, after the zebrafish were immersed in a high-glucose solution for 14 days, the blood glucose levels of zebrafish were continuously increased, the response to exogenous insulin was weakened, and the mRNA level of insulin receptors in muscle tissue was decreased. After treatment with glimepiride and metformin, the high glucose metabolism was improved. When the zebrafish were continuously exposed to a sugar-containing solution, blood glucose levels also increased, but this increase was transient. Moreover, continuous immersion in a sugar solution, especially at high concentrations, can lead to abnormal swimming and gill function in zebrafish, and some zebrafish died during the modeling process [85,86,87].

#### 2.2.2. High-Fat Food Induction Method

Obesity is one of the inducing factors of T2DM in mammals. It selectively destroys the satiety center of the ventromedial hypothalamic nucleus to cause animals to binge eat. A special laboratory diet is used to induce obesity in animal models; this is followed by hyperglycemia, hyperinsulinemia, and insulin resistance, which are similar to human NIDDM. Both classic and alternative NF-κB signaling pathways have been found to be associated with metabolic syndrome, insulin resistance, and β-cell dysfunction in diabetes. NF-κB-inducible kinase (NIK) is elevated in the muscle tissues of obese patients. This indicates that the NF-κB alternative signaling pathway is activated [88,89]. Zang et al. established a zebrafish model of T2DM by feeding three-month-old adult zebrafish six times the normal food intake to induce obesity. Intraperitoneal glucose tolerance and oral glucose tolerance tests showed that overfeeding resulted in impaired glucose tolerance and that hyperglycemia was significantly relieved by treatment with glucose-lowering drugs. These phenotypes were similar to the insulin resistance phenotype of type 2 diabetes. In addition, Nishimura’s team achieved diet-induced obesity (DIO) in zebrafish by overfeeding. Enhanced green fluorescent protein (EGFP) labeling of insulin in DIO zebrafish was performed to obtain *Tg* (*ins:EGFP;DIO*) zebrafish. The zebrafish were overfed for 8 weeks; each fish was fed 120 mg per day by an automatic feeder. Body weight and fasting blood glucose levels were measured weekly for 8 weeks. After overfeeding, the glucose tolerance and insulin content (indirectly reflected by green fluorescent protein) of the zebrafish were detected, and abnormal glucose metabolisms were found. This finding is similar to the abnormal glucose metabolism observed in the spontaneous remission type 2 diabetic mouse model established by a study [90,91,92,93,94,95,96,97,98]. Although high-fat feeding is simple to perform, it takes longer. Additionally, obesity often leads to cardiovascular disease and endocrine abnormalities.

#### 2.2.3. CRISPR/Cas9 Gene Knockout Method

CRISPR/*Cas*9 is a revolutionary gene editing technology that was developed in recent years, and its application in zebrafish research is well established. The application of CRISPR/*Cas*9 technology to inject guide RNA (gRNA) and *Cas*9 mRNA at the single-cell embryonic stage can quickly obtain gene knockout zebrafish, which provides an ideal model for the study of pancreatic diseases and glucose metabolism. It is an effective technical means to construct diabetic zebrafish models [99,100,101,102].

Insulin resistance is mainly caused by decreased sensitivity to insulin and its target organs. Therefore, knocking down the expression of the insulin receptor (*INSR*) gene in target organs can lead to increased blood glucose levels. Maddison et al. generated transgenic zebrafish with muscle *INSR* gene knockdown in which insulin-resistant skeletal muscle and the insulin-like growth factor-I (IGF-I) receptor were inactivated in the skeletal muscle of zMIR transgenic zebrafish. Initially, these zebrafish showed a compensatory increase in the number of β-cells and normal glucose tolerance. However, with increasing age, the number of β-cells decreased, glucose tolerance was impaired, and fasting blood glucose levels increased. There are two types of *INSR* in zebrafish [103,104]. Gong et al. and Yang et al. knocked out the *INSR* gene in these two types of zebrafish, and the results showed that zebrafish with *INSR* gene knockout showed hyperglycemia and a compensatory increase in β cells. *insra*^−/−^ zebrafish had much higher postprandial glucose level elevation than *insrb*^−/−^ zebrafish. *insra*^−/−^ zebrafish had a disorder that inhibited gluconeogenesis and stimulated glycolysis, and *insrb*^−/−^ zebrafish also had a serious defect that stimulated glycolysis. These results suggest that *insra* and *insrb* have overlapping and diverse functions in maintaining blood glucose homeostasis in zebrafish. However, MKR transgenic type 2 diabetic mice with low *insra* expression in muscle tissue showed hyperglycemic symptoms, but no compensatory increase in β cell mass was observed. This might be caused by impaired glycogen storage and impaired gluconeogenesis inhibition in *insra*^−/−^ and *insrb*^−/−^ zebrafish [105]. Luis et al. developed a zebrafish model of IL-1β-mediated islet inflammation, *Tg* (*ins:il1b*). Some amino acids of IL-1β were deleted to simulate the activated form cleaved by caspase-1. The subsequent inflammatory response specifically damaged the function and characteristics of β cells. They showed a significant reduction in calcium influx after glucose stimulation; a decrease in the expression of genes related to β-cell maturation and function, such as the transcription factor *Pdx1* and the type 2 diabetes risk gene *kcnj11*; impaired glucose tolerance, and hyperglycemia [106]. At present, the use of CRISPR/*Cas*9 to construct diabetic zebrafish models also faces systematic defects, such as off-target genes and genetic instability after gene editing.

#### 2.2.4. Genetic Ablation Method

Conditional targeted ablation is a powerful tool to study the roles of specific cell lines or tissues in developmental or physiological processes, and a wide range of genetic cell ablation techniques have been developed and used in different model systems. For example, Lewandoski used gene ablation to achieve conditional control of gene expression in mice, but in zebrafish, no genetic cell ablation tool has been shown to be spatially controllable, strictly limited to the target cell population, and temporally inducible. Curado et al. then used a bacterial nitroreductase (NTR) to convert the prodrug metronidazole (MTZ) into a cytotoxic DNA crosslinker, and found that by combining chemical and genetic tools, one could ablate cells in zebrafish larvae in a temporally and spatially specific manner. By combining chemical and genetic tools, one can ablate the cells of zebrafish larvae in a temporally and spatially specific manner [107,108].

In summary, each method for constructing diabetic zebrafish models has its own advantages and disadvantages, and no one method exactly meets the needs of all research projects thus far (Table 2). Therefore, researchers need to either choose the appropriate construction method according to their own research needs or combine two or more methods to construct an ideal diabetes model for the mechanistic study of diabetes and its complications, or for the high-throughput screening of related therapeutic drugs.

### 2.3. Evaluation Indicators of Diabetic Zebrafish Models

The most typical feature of diabetes is persistent hyperglycemia. The blood glucose level is the main indicator of pancreatic β-cell function, and it is also a commonly used indicator of a clinical diagnosis of diabetes and is used in in animal models of diabetes. To study glucose metabolism in zebrafish, it is common practice to collect the blood of adult zebrafish after anesthesia by detecting indicators, such as fasting blood glucose levels and postprandial blood glucose levels, and performing a glucose challenge. Under normal physiological conditions, the blood glucose level of zebrafish is in a state of dynamic self-regulation. When fasted for 2 days, the blood glucose level will increase in response to stress, and it will drop to the normal level on the third day. The blood glucose level increases and peaks within 30 min after glucose injection and returns to normal levels at 6 h after glucose injection. In the blood glucose detection of zebrafish, the anesthesia method is a factor affecting the blood glucose value. Most anesthetics are nerve cell ion channel blockers, which directly cause the β cell channel to affect insulin secretion. The anesthetic MS-222 has been reported to interfere with β-cell function and cause blood glucose elevation in teleost fish. It has been found that the CV value of zebrafish blood glucose in the MS-222 anesthesia group was increased, so hypothermia anesthesia was used in his study [118,119]. In some vertebrates, including teleost fish, blood glucose levels are also affected by sex, but no sex difference was detected in zebrafish blood glucose levels in the experiment. Zebrafish larvae are small in size and have a small amount of blood, so it is impossible to measure blood glucose values by collecting blood [120]. Li et al. used the “protein quantification method” to measure blood glucose values indirectly by using zebrafish larvae homogenate [121]. Due to the small size of zebrafish, it is difficult to collect large enough blood samples. Therefore, it is still necessary to find a simpler and more effective method to evaluate islet function for the study of zebrafish pancreatic function and glucose metabolism.

## 3. Types of Zebrafish Models of Diabetes and Their Complications

### 3.1. Maturity-Onset Diabetes Mellitus Model

Maturity-onset diabetes mellitus (MODY) of the young is a type of diabetes caused by a single gene mutation of a genetic defect in pancreatic β-cell function. It is characterized by an early age of onset and autosomal dominant inheritance. The homologous mutation of variant hepatocyte nuclear factor 1 (vHnf1) is associated with MODY5. The vHnf1 zebrafish mutant showed pancreatic and hepatic hypoplasia, renal cysts, and renal fibrosis. However, mutants with severe loss of vHnf1 function or with local defects in the foregut and pancreatic hypoplasia limit its usefulness as a model for studying pancreatic function. In the novel *hnf1ba* zebrafish mutant, *hnf1ba* function is partially lost, resulting in MODY5-like pancreatic hypoplasia, reduced β-cell numbers, and no apparent regional defect in the foregut endoderm [122,123,124].

### 3.2. Gestational Diabetes Mellitus Model

Gestational diabetes mellitus (GDM) has a profound effect on fetal development. Singh et al. used the pulsed high glucose exposure method to expose wild-type zebrafish embryos to high glucose every 24 h to simulate the changes in the intrauterine blood glucose levels that occur in mothers with GDM. Compared with the control group, the total free glucose level of zebrafish embryos in the high glucose group fluctuated in a dose-dependent manner, and the mean free glucose level was significantly higher. Notably, the high glucose group had obvious defects in retinal development, the thickness of the retinal cell layer was changed, and the number of Mueller glial cells and retinal ganglion cells was decreased [125,126]. A previous study successfully established a juvenile zebrafish model of type II diabetes mellitus by continuously treating zebrafish embryos with an embryo culture medium containing a 1.5% glucose solution. The results showed that sustained hyperglycemia caused severe oxidative stress and severe developmental toxicity in juvenile zebrafish.

### 3.3. Diabetic Cardiovascular Disease Model

Hyperglycemia is an independent risk factor for heart damage, which can lead to diabetic cardiomyopathy. Zebrafish are widely used as an animal model in the study of cardiovascular diseases. Sun et al. successfully established an adult zebrafish model of diabetic cardiomyopathy, and zebrafish treated with high glucose gradually developed myocardial hypertrophy, apoptosis, and arrhythmia. Echocardiography showed early diastolic dysfunction and late systolic dysfunction of the heart, which was consistent with observations in patients with diabetes. These findings confirm that high glucose levels can induce cardiac remodeling and dysfunction in adult zebrafish. Transgenic and gene-knockout constructs expressing green fluorescent protein (Fli1:EGFP) in endothelial cells, green fluorescent protein (Lyz: EGFP) in macrophages, and red fluorescent protein (Gata1: vascular fluorescent zebrafish (dsRed) have been widely used in cardiovascular research, and the establishment of a diabetic zebrafish model using vascular fluorescent zebrafish greatly improved the efficiency of the research [127,128,129,130].

### 3.4. Diabetic Retinopathy Model

Diabetic retinopathy (DR) is one of the common microvascular complications of diabetes mellitus and the main cause of blindness in patients with diabetes. The visual system of zebrafish is very similar to that of humans and other mammals. In zebrafish, the entire cell layer of the optic reticulum can be clearly observed, from the ganglion cell layer to the retinal pigment epithelium layer. Zebrafish have been widely used in the study of visual development and restoration. The inner plexiform layer (IPL) and inner nuclear layer (INL) of zebrafish retinas exposed to high glucose solutions were both decreased compared with those in the normal group, and the IPL was significantly thinner, similarly to other diabetic animal models and patients with diabetes [131,132]. Jung et al. treated zebrafish embryos with high glucose concentrations and observed vitreoretinal vascular dilatation with morphological damage, disruption of tight junction proteins, and an increase in vascular endothelial growth factor (VEGF) mRNA and NO on the sixth day. This short-term diabetic retinopathy model will be an effective tool for screening potential therapeutic drugs for DR [133]. Recently, it was found that the transcription factor Pdx1 was knocked down using CRISPR/*Cas*9 technology. *Pdx*1^−/−^ mutants showed impaired pancreatic development and hyperglycemia, as well as dilated blood vessels and increased permeability in larvae. The early vascular phenotype induced by hyperglycemia can persist into adulthood, and vascular changes in the retinal vasculature are exacerbated, leading to vascular hyperbranching [117]. VEGF and NO inhibitors and hypoglycemic agents alleviate the vascular changes caused by hyperglycemia, but the higher mortality of *pdx*1^−/−^ mutants limits their use in adulthood. In DR, hyperglycemia can lead to retinal microvascular lesions, and retinal damage can stimulate the growth of new blood vessels to form proliferative retinopathy. Therefore, the observation of retinal vascular morphology is essential in the study of diabetes [134]. At present, some specific zebrafish angiogenesis models have been established which can be used for the observation and study of retinal vessels in DR. However, the retinal vascular system of zebrafish is relatively complex, and the retinal venous system is also different from that of humans. Therefore, when using zebrafish for DR research, attention should be given to possible deviations at the cellular or vascular level.

### 3.5. Neurological Complications

Zebrafish are used as a common animal model in neurobiology research. The response of zebrafish to drugs is highly similar to that of humans. In addition, zebrafish also possess important neurotransmitter transmission systems, including cholinergic, 5-hydroxytryptaminergic, dopaminergic, and noradrenergic systems. Researchers have also found that zebrafish are similar to humans in terms of learning, sleep, drug addiction, and other neurobehavioral phenotypes. Many genes associated with human nervous system diseases are highly conserved in zebrafish, which indicates that zebrafish can be used for modeling neurological diseases [135,136,137]. Dorsemans et al. developed a new model of acute hyperglycemia to explore the effects of acute and chronic hyperglycemia on brain homeostasis and nerve regeneration. Studies have shown that acute hyperglycemia increases the expression of proinflammatory cytokines in the brain, whereas chronic hyperglycemia decreases the expression of genes involved in brain cell proliferation and blood-brain barrier formation [138]. In addition, hyperglycemia can regulate acetylcholinesterase (ACHE) function and gene expression in zebrafish, leading to choline dysfunction and, eventually, memory loss. Galantamine can reverse hyperglycemic-induced memory deficits. Rocker et al. verified the effect of hyperglycemia on the peripheral nervous system of zebrafish through NTR/MTZ-mediated β cell elimination, reductions in motor neurons, disruptions in tight junctions, and alterations in Schwann cell numbers in hyperglycemic juvenile fish [139,140].

### 3.6. Diabetic Nephropathy

Diabetic nephropathy (DN) is a chronic disease caused by diabetes mellitus. DN is also a common complication of diabetes and the main cause of end-stage renal disease. Patients presented with tubular basement membrane thickening, glomerular membrane thickening, or extracellular matrix accumulation. STZ-induced type 1 diabetic zebrafish showed glomerular basement membrane (GBM) thickening at one week [141]. One study found that knockdown of *Pdx1* by antisense morpholino (MO) technology induced hyperglycemia in zebrafish, which resulted in pronephric glomerular dilatation and high atrophy of the filtration barrier. On the basis of the hyperglycemic zebrafish model, knocking down *Pdx*1 and erythropoietin (EPO) simultaneously aggravated kidney injury, further increased glomerular length, and significantly shortened the pronephros neck. These findings reveal the protective effects of EPO against diabetic nephropathy. The pancreatic duodenal homeobox factor 1 (*Pdx* 1) gene plays an important role in the formation and differentiation of the zebrafish pancreas. Zebrafish embryos established via *Pdx* 1 gene knockdown by morpholine generation showed hyperglycemia, which resulted in the enlargement of the anterior glomerulus, impairment of the anterior renal filtration barrier, and defective podocyte development. Moreover, hyperglycemia induced by intraperitoneal injection of streptozotocin in adult zebrafish led to glomerular basement membrane thickening, which is also observed in humans [142,143,144].

### 3.7. Diabetic Wound Model

Diabetes mellitus (DM) is known to cause poor wound healing. In Michael’s study, DM zebrafish also exhibited impaired fin regeneration and skin wound healing, which persisted even after the blood glucose levels returned to normal. This finding is consistent with the clinical manifestations of diabetes. That is, once diabetes occurs, even if the blood glucose level is controlled to normal levels by medical intervention, some complications will still exist and continue to progress. Several other experiments have also demonstrated that the initial hyperglycemic phase leads to long-term dysfunction of target organs. This phenomenon has been named the metabolic memory (MM) effect. Michael further studied the expression of genes in the two states of hyperglycemia and blood glucose recovery and found that the expression patterns of genes in the two states were similar. This was the reason for the occurrence of fin and skin aplasia even after the blood glucose level and other indicators returned to normal. The damage caused by hyperglycemia to target organs remains irreversible [111,145].

### 3.8. Diabetic Immune Injury Model

Patients with diabetes have varying degrees of immune damage, which involves abnormalities in cellular immunity, humoral immunity, red blood cell immunity, and other immune damage parameters. There are two main reasons. First, long-term hyperglycemia itself will cause the body to be in a chronic inflammatory state for a long time, which directly leads to low immunity and immune disorders. Second, diabetes and impaired immune functions share a common pathogenetic basis. Therefore, diabetes and impaired immune functions can affect each other. T1DM is characterized by the T lymphocyte-mediated destruction of insulin-producing islet β cells. The pathogenesis of type 2 diabetes involves β-cell loss and persistent endoplasmic reticulum stress. “Lisette A. Maddison et al. constructed a zebrafish skeletal muscle insulin resistance (zMIR) model to study the cellular mechanisms of β-cell loss and the complex intercellular communication involving macrophages, neutrophils, and stressed β-cells themselves.” We found that ER stress led to increased numbers of macrophages and neutrophils in the zMIR model, in which macrophages were able to induce TNF-α dependent *Cxcl8a* expression in β cells. *Cxcl8a* then recruited neutrophils, and neutrophils attacked β cells in contact with macrophages, resulting in their loss. This demonstrates the effect of diabetes on the function of the immune system. It was also found that the depletion of macrophages and neutrophils could alleviate β-cell damage and the symptoms of hyperglycemia [146,147,148], which provides new opportunities for advancement in the study of diabetes (Table 3).

Therefore, diabetes mellitus not only increases blood glucose levels, but also affects whole-body metabolism due to the long-term maintenance of high blood glucose. Thus, cardiovascular and cerebrovascular diseases, chronic damage to various functional tissues (especially the eye, kidney, heart, blood vessels, and nerves), dysfunction complications, and even death are induced. Therefore, the establishment of different zebrafish models of diabetes and its complications is necessary to identify the causes and results of diabetes and its complications and to effectively alleviate the symptoms of diabetes patients. In summary, the construction of a zebrafish animal model is of great significance not only for the study of the pathogenesis of diabetes, but also for the study of the pharmacological effects of related therapeutic drugs.

## 4. Conclusions and Prospects

The establishment of a good diabetic animal model is beneficial to the study of diabetic pathological mechanisms and the development and screening of clinical drugs. This review summarizes the construction methods of diabetic zebrafish. In particular, the construction of type 1 diabetes models mainly occurs through surgical resection of the pancreas, induction with drugs such as streptozotocin or alloxan, and induction with gene modifications. The construction of a type 2 diabetes model is mainly achieved by soaking zebrafish in glucose solutions, inducing obesity by feeding a high-fat diet, and using CRISPR/*Cas*9 gene knockout technology to complete the model’s construction. Each of these methods has its own advantages and disadvantages. Researchers should choose according to their specific situation or improve and innovate on the basis of the existing methods. In addition, in this review, several models of zebrafish diabetes and its complications, such as the juvenile diabetes model, gestational diabetes model, and diabetic retinopathy model, as well as the research status of these models in diabetes research, were also described (Figure 3).

The establishment of a good diabetes animal model is conducive to the study of the pathological mechanism of diabetes as well as the development and screening of clinical drugs. Zebrafish have a unique advantage that rodents do not have. Genetic manipulations are easy, and the establishment of the models is fast and efficient. Furthermore, these models can be used to accurately analyze the pathogenesis of diseases at the molecular level. This could effectively bridge the gap between in vitro cell experiments and traditional rodent experiments. The genetic background of zebrafish is well-defined. Thus, zebrafish could serve as a new diabetes model for studying the pathogenesis of diabetes, and the relevant research results may provide a new theoretical basis for the treatment of human diabetes. Therefore, the study of the mechanisms of diabetes in zebrafish can provide a good experimental basis for the clinical treatment of diabetes.

In summary, zebrafish have unique advantages that rodents do not have. Transgenic methods are easy to perform, and animal models can be built rapidly and efficiently. This model can be used to accurately analyze the pathogenesis of diseases at the molecular level and effectively bridge the gap in biological experiments between in vitro cell experiments and traditional rodent experiments. The genetic background of zebrafish is well-defined. Thus, zebrafish could serve as a new diabetes model for the study of the pathogenesis of diabetes, and the relevant research results may provide a new theoretical basis for the treatment of human diabetes. Therefore, the study of the mechanisms of diabetes in zebrafish can provide a good experimental basis for the clinical treatment of diabetes. In addition, the zebrafish diabetes model can compensate for the deficiencies of other diabetic animal models. The structure, function, and regulation of glucose metabolism in the zebrafish pancreas are highly consistent with those of mammals, and this model has been widely used in research on pancreatic developmental biology, the study of different types of diabetes and its complications, the screening of diabetes drugs and drug toxicity, and other fields. New and diversified diabetic zebrafish models are still under construction and will play an increasingly important role in the study of diabetic diseases.

At present, there is no radical cure for diabetes. Diabetes can cause damage to many organs during its onset, and the etiology of complications of the disease is highly complex. Although zebrafish share 87% of homologous genes with humans, they are not exactly the same as humans, so there may be uncertainties in the study that still need to be perfected. However, the construction of a variety of diabetic zebrafish models has played a significant role in the study of diabetic diseases. In the future, through the continuous construction and research of a variety of diabetic zebrafish models, in vivo imaging in zebrafish, and chemical genetic manipulations, we hope to solve the difficulties related to diabetes and its complications and make a significant contribution to finding a radical cure for diabetes.

## Figures and Tables

**Figure 1 ijms-24-05195-f001:**
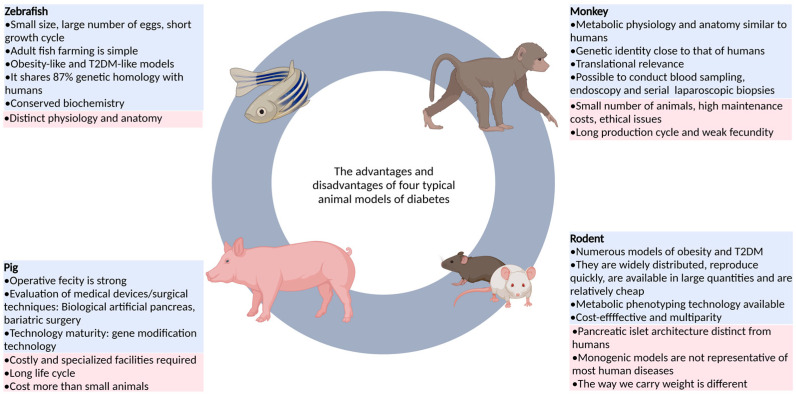
The advantages and disadvantages of four typical animal models of diabetes.

**Figure 2 ijms-24-05195-f002:**
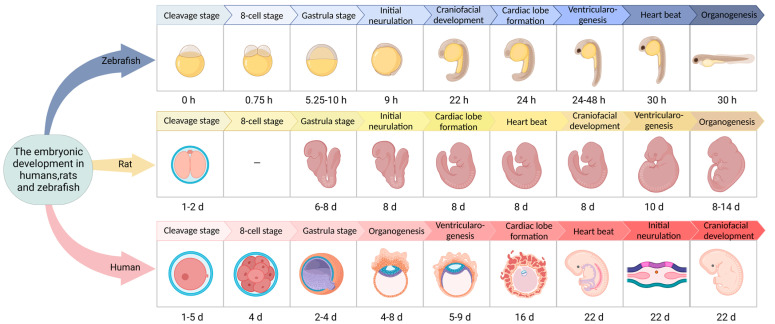
Embryonic development in humans, rats, and zebrafish.

**Figure 3 ijms-24-05195-f003:**
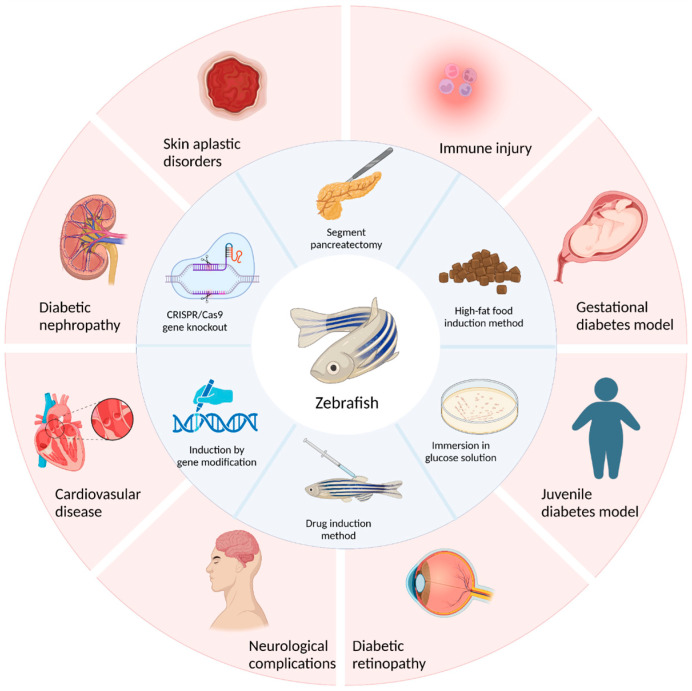
Study on the construction and application of the diabetic zebrafish model.

**Table 1 ijms-24-05195-t001:** Advantages and disadvantages of several animal models of diabetes mellitus.

Category	Animal	Advantage	Disadvantage	References
Non-mammalian animals	Zebrafish	As an emerging model, it has the advantages of fast reproduction, copulatory behavior controlled by photoperiod, large number of eggs laid, in vitro fertilization of fertilized eggs, overall transparency of early embryos, easy feeding, and easy use of drugs.	It is an ectothermic animal and lacks brown adipose tissue, so it is still difficult to measure insulin levels and assess insulin resistance.	[15,16,21]
	Drosophila melanogaster	Drosophila is easy to obtain and operate, and has strong conservation with the lipid-related metabolism genes of mammals, which can be used to study the function of candidate genes related to T2DM.	It is distantly related to humans, and its anatomical structure and physiological function are slightly similar to those of the human body.	[22,23]
Rodent	Rat and mouse	It is highly productive, low-cost, and diverse, covering obesity and diabetes, with advanced tools for genetic modification and metabolic phenotype assessment.	The structure of islets, basal metabolic rate, feeding behavior, immune system, and gut microbiota are less similar to those of humans.	[24,25,26]
Large mammals	Pig	The feeding cost is low and wide application is possible. The gene modification technology is relatively mature, and a variety of disease models can be generated through targeted gene editing.	It does not usually lead to diabetes and requires a combination of other means. It also requires special equipment and costs more than small animals.	[13,27,28]
Non-human primates	Monkey	The anatomical structure, physiological characteristics, and genetic background are similar to those of humans, and the results of the study are of high clinical relevance.	The number of animals is small, the production cycle is long, the fecundity is weak, the technology is imperfect, and the price is high.	[29]

**Table 2 ijms-24-05195-t002:** The construction methods of zebrafish diabetes models and their advantages and disadvantages.

Type	Construction Method	Advantages	Disadvantages	References
Type I Diabetes	Surgical resection method	The earliest method for replication of animal models of diabetes.	It requires highly skilled operating techniques and elaborate equipment for researchers, and the trauma to zebrafish is significant, resulting in a low survival rate of zebrafish after surgery.	[109,110]
	Drug induction method	With the advantages of short time frames, simplicity, ease of mastery, and good repeatability, a large number of models can be induced in a short period of time.	Multiple intraperitoneal injections of streptozotocin are needed, and the operation is complicated.	[111,112,113]
	Induction by genetic modification	Transgenic technology can be used to construct zebrafish with specific functions.	Transgenes will damage other genes in the genome, resulting in the loss of other functional genes, and the acquisition of homozygotes takes a long time.	[108,114]
	Genetic ablation method	Target cell populations in zebrafish larvae can be effectively removed by the NTR/MTZ system.	This approach has limitations due to the instability and uncertainty of current techniques regarding the genetic aspects.	[108,109,115]
Type II Diabetes	Glucose solution immersion method	The feeding procedure is simple and the diabetic model can be established in about ten days.	In solutions with high concentrations of sugar, the swimming and gill functions of the zebrafish were abnormal, and some zebrafish died in the modeling process.	[116]
	High-fat food induction method	The feeding procedure is simple.	It takes a long time, and obesity often leads to cardiovascular disease and endocrine abnormalities.	[90]
	CRISPR/Cas9 gene knockout method	Knockout zebrafish can be obtained quickly.	In the face of gene editing, the target gene is off-target, and there is genetic instability after editing.	[117]

**Table 3 ijms-24-05195-t003:** The types, construction methods, and phenotypic characteristics of zebrafish models of diabetic complications.

Type of Diabetic Zebrafish Model	Construction Method	Phenotypic Characteristics	References
Juvenile diabetes model	Mutation of hepatic nuclear factor 1β(*HNF-1β*) gene	Early onset, autosomal dominant inheritance, diabetes mellitus, renal cysts (dysplastic glomerular cystic lesions), and even nondiabetic renal insufficiency.	[149,150]
Gestational diabetes model	Glucose immersion method	The total free glucose level of zebrafish embryos in high glucose group fluctuated in a dose-dependent manner, and the average free glucose level increased significantly.	[151]
Diabetic cardiovascular disease model	Glucose immersion method	Zebrafish treated with high glucose concentrations gradually showed cardiac hypertrophy, apoptosis, and arrhythmia. Early diastolic dysfunction and late systolic dysfunction occurred in the heart.	[113]
Diabetic retinopathy model	Glucose immersion method	The internal layer and core layer of zebrafish in the diabetes model group were significantly lower than those in the normal group, and the IPL was significantly thinner.	[100,109]
Neurological complications	Glucose immersion method	Hyperglycemia can regulate acetylcholinesterase function and gene expression in zebrafish, leading to choline dysfunction and eventual memory loss.	[121,123,137]
Renal complications	CRISPR/*Cas*9 gene knockout (CRISPR/*Cas*9-induced ELMO1 deletion in 48hpf embryos)	The basal glomerular membrane was thickened.	[128,138]
Diabetic wound model	Streptozotocin drug induction method	In diabetic zebrafish, fin regeneration and skin wound healing are impaired and persist even after blood sugar levels return to normal.	[152,153]
Diabetic immune damage model	The dominant negative mutant of insulin-like growth factor 1 receptor was constructed by using the dominant negative effect.	In this model, ER stress led to an increase in the number of macrophages and neutrophils, and neutrophils attacked beta cells in contact with the macrophages, causing them to be lost.	[91]

## Data Availability

Not applicable.

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
