# Peer review of "Research Progress on the Construction and Application of a Diabetic Zebrafish Model"

_ijms, 2023, doi:10.3390/ijms24065195_

Round 1
Reviewer 1 Report
Dear Authors,
I was in charge of the review. It was a very interesting manuscript. The subject of this study discusses the feasibility of using zebrafish as an animal model for diabetes. Many researchers do not know about this. We hope that this report will serve as a reference for many researchers. I have some suggestions to make this report better. I have listed them below.
1. Page 2, line 45 "Ammons Seeds Studies..." and line 56 "Ammons Seeds Studies..." are repeated. The former does not include "mice". Must include mice. I need to revisit the entire sentence.
2. Shouldn't Figure 1 say "Rat and mouse" or "Rodent" instead of "Rat"?
3. Page 3, lines 111-113 Regarding "Their carbohydrate~", do you have any references?
4. Regarding "Reference" in Tables 1 and 2, why don't they appear in the text? Also, why is the document introduced in the text not listed in this "Reference"?
I think the conclusion is consistent with the discussion.
Reviewer 2 Report
This is potentially interesting review for scientific community but needs more work. There is a lot repetitions in the text- not only expressions or sentences but also a major parts of some paragraphs! It has to be removed eg. 2, 3 and 4th they are almost similar. Line 68 and 77: the same! I would rather say that: „the cellular structure of zebrafish is similar to…” not opposite- zebrafish might be similar to human but not human to zebrafish. Line 99: again repetition with the same words.
Line 106: embryos are exposed to drugs, but not drugs to embryos
Line 111: i dont know what Authors meant here, for re-writing. Also, why 3 times in one sentence „carnivorous”?
Line 117: i ve never heard about expression that „gene is actively expressed”, gene is expressed in enough
Line 135: „targets damaged beta cells” for re-writing
Line 137: remove „induction” for gene modification
Line 239: rather: „Special laboratory diet is used to induce obesity in animal models”
Reviewer 3 Report
The review article is written nicely. The information provided will be very helpful in understanding the current scenarios in the zebrafish model and its application in diabetes research. Various challenges have been highlighted, and these will help to further improve the model. Overall the review has important implications. I have certain concerns, as mentioned in the comment below.
Comments.
1. The authors repeatedly mentioned that “zebrafish has many advantages, such as its small size, a large number of eggs, short growth cycle, simple cultivation of adult fish, and mentioned that this model is highly suitable for research as an animal model of diabetes. How these features are helpful in diabetes research is key, and the authors should emphasize these. The author should cite some link between these features and the disease.
2. Figure 1 has been well presented. However, the review article is focused on diabetes, and the authors should describe these characteristics from the disease's perspective. Which features will be important in the following model organisms which are key to diabetes? Elaborating on each characteristic to include their implication in diabetes would be helpful.
3. A table describing the various models available so far and their references would be helpful.
4. In the introduction, line 48 authors say, “Adult fish culture is simple and can effectively improve experimental efficiencies. Thus, this animal model is very suitable for human disease modeling and mechanical research, and this is a very vague explanation. Other important features, such as homology in diabetic genes, organs, and available transgenic specific to the disease, would be helpful.
5. Please provide a reference for the paragraph (Line 41-52).
6. Lines 41-43 and 53 to 54 are repetitions. Please check.
7. The author wrote, “In the field of scientific research, the zebrafish model has gradually attracted the attention of many researchers due to its many advantages that other model animals do not have, and it has become a very important vertebrate model organism (Figure 1). Here as mentioned in Comment 3, a table of comparisons (a feature important in diabetes) would be helpful. Importantly, references should be added to each statement in the figure/table.
8. The authors wrote,” In the field of scientific research, the zebrafish model has gradually attracted the attention of many researchers due to its many advantages that other model animals do not have” Further, they wrote, “has become a very important vertebrate model organism (Figure 1). For example, zebrafish are easy to breed because of their small size, large number of eggs, short growth cycle, and simple adult fish breeding, which can effectively improve experimental efficiency. Key advantages are missing, which include the optical clarity of the larvae, which makes them amenable to accurate time tracking of the molecular processes, including organ development. Again, the reference in this para is missing.
9. The authors wrote (Lines 69-71 “The zebrafish model has its own characteristics and advantages in terms of endocrine gland function and glucose metabolism, and this model has been applied to studying human diabetes and other metabolic diseases. Here a reference is important.
10. The author wrote “Different developmental stages of zebrafish can be found in the development process of mammals. Furthermore, zebrafish only need 72 hours to develop from fertilized eggs to hatching. The authors should include more information about the stages of development in the text, which resembles mammals, as shown in the figure. Also, why this hatching stage is crucial? For example, in zebrafish, most major organs are developed within 72 hours.
11. The authors concluded that “This review summarized the construction methods of diabetic zebrafish”. However, there are some information missing. For instance, in table 1. For Type 1 D genetic ablation methods are missing (Curado et al., 2007, 2008, Li et la., 2014. Delaspre et al., 2015.
12. It will be helpful to discuss various transgenic zebrafish models available to study the disease. For instance, Zhang et al 2017 and Kinkel and Prince, 2009; Tiso et al., 2009; Prince et al. etc.
13. How well is the biology in glucose homeostasis, is conserved from zebrafish to humans at the level of genes to organ? The authors should discuss the robust application of the zebrafish model would be the live imaging in zebrafish, coupled with genetic and chemical genetic manipulations, which will likely yield insights to many outstanding questions in diabetes.
Round 2
Reviewer 3 Report
Thank you for addressing the previous concerns by responding to each comment suitably and adequately. The review has improved significantly.